# Access to healthcare among transgender and non-binary youth in Sweden and Spain: A qualitative analysis and comparison

María del Mar Pastor Bravo[1,2☯], Ida Linander[3☯] *

1 Department of Nursing, University of Murcia, Murcia, Spain, 2 Biomedical Research Institute of Murcia (IMIB) Pascual Parrilla, Murcia, Spain, 3 Department of Epidemiology and Global Health, Umeå University, Umeå, Sweden

☯ These authors contributed equally to this work.
* ida.linander@umu.se

**Data Availability Statement:** The interview data contain identifying and sensitive information. Even if identifying information such as name, geographical location and occupation would be

## Abstract

### Background

Transgender and non-binary (TGNB) people tend to report worse health than cis people, however, despite an increased need for care, they face several barriers when trying to access healthcare. These barriers might be exacerbated when young age intersects with a trans identity, and so there is a need for studies highlighting the experiences of TGNB youth.

### Aims

To explore and compare how TGNB youth (15–26 years old) in Sweden and Spain experienced their access to healthcare, in order to shed light on the strengths and limitations of different kinds of healthcare systems and improve healthcare provision and policy development.

### Methods

This study was based on a qualitative analysis of semi-structured interviews with TGNB youth living in Sweden (n = 16) and Spain (n = 18). Of these, 22 identified as male or transmasculine, six as non-binary, and six as women or transfeminine; 25 had undergone some type of gender-affirming care, and the rest were on the waiting list or undergoing preparatory visits and had not started hormonal treatment. The interviews were analyzed using reflexive thematic analysis. An abductive approach was applied, and the Levesque conceptual framework was used to compare the analyses of each set of materials.

### Results

We present our findings using the structure of the accessibility framework, focusing on approachability, acceptability, availability, affordability, and appropriateness. The conceptualization of accessibility in combination with the concept of cisnormativity illustrates how specific ideals and normative expectations affect access to healthcare for TGNB people across contexts, with most barriers arising from the appropriateness of the services.

removed, the combination of information in the interviews would potentially compromise the anonymity of the participants. The participants also belong to a small community, which increases the risk of identification of participants. Access might be granted after an ethical review. For questions/requests contact the Swedish Ethical Review Authority (Etikprövningsmyndigheten, Box 2110, 750 02 Uppsala, registrator@etikprovning.se) or Research Ethics Commission (Comision de Etica de Investigacion, Universidad de Mucia, Spain, comision.etica.investigacion@um.es).

**Funding:** The work in Sweden work was supported by FORTE (grant number 2019:00355). The funders had no role in study design, data collection and analysis, decision to publish, or preparation of the manuscript. The work in Spain did not receive any specific funding for this work.

**Competing interests:** The authors have declared that no competing interests exist.

## Discussion

Young TGNB people experience barriers to accessing healthcare both in the Spanish and the Swedish contexts. Strategies to reduce these barriers should be framed within the critique of and resistance to cisnormativity and should focus on users with intersecting marginalized identities to promote health equity.

## Introduction

Healthcare systems are an important determinant of health and can contribute to more equitable levels of health by providing more care for those in greater need [1,2]. However, although transgender and non-binary (TGNB) people tend to report worse health than cis people, especially regarding mental health, studies have shown that trans people face several different barriers when trying to access healthcare [3–9]. These barriers can be exacerbated when other axes of vulnerability intersect with a trans identity, such as age [10,11].

Barriers that have been identified include experiences of stigma, prejudice, discrimination, financial barriers, and a lack of knowledge and visibility when seeking healthcare [3,5,6,8,9,12,13]. Studies have also shown that TGNB people have experiences of having to educate their healthcare providers, having their gender identity questioned, being asked irrelevant questions, and getting comments and questions about body parts [13–15]. Furthermore, non-binary trans youth have experiences of being binary-gendered in healthcare settings and finding more barriers to accessing gender-affirming healthcare [16]. Quantitative studies have also suggested that TGNC youth might have limited access to preventive care, but also might visit the school nurse more than cisgender youth [17].

Many TGNB people postpone healthcare visits even if they are in need of them [4,5,8,13]. A report by Transgender Europe [8] showed that 62% of the Swedish participants and 48% of the Spanish participants had delayed visits to a general practitioner, with those who had previous bad experiences of healthcare being more inclined to postpone healthcare visits.

Several studies have pointed out that access to gender-affirming healthcare, such as hormone treatment and breast surgery, is limited in different ways. Some of these barriers also apply to general healthcare; for example, lack of knowledge [12]. In some national contexts, access to gender-affirming care is limited due to a lack of coverage by healthcare insurance, which means that care-seekers are dependent on their own financial means to access vital care [9]. In other countries, such as Sweden, Denmark, Germany, France, Finland, the Netherlands, the United Kingdom, and Spain, most gender-affirming medical procedures are subsidized for citizens or people with permanent residency [9].

However, research has shown that access to gender-affirming healthcare is limited in other ways, independent of the financial system. One specific barrier is comprised of demands for mental health evaluation and the gatekeeping role of mental healthcare providers; that is, their control over access to medical procedures. Pathologization in the form of psychiatric diagnoses has been reported to be experienced as stigmatizing and hindering access [18]. There are also specific issues regarding TGNB youth access to gender-affirming care, due to policies being restrictive with regard to both hormones and surgery, especially for people aged under 18 (see, for example [19]).

Although there is an emergent body of literature exploring TGNB people's access to care, there is a lack of studies focusing on the specific experiences of young people and comparing trans people's access to care across national contexts. Qualitative studies that highlight the

voices of young TGNB people are needed, as this is a group that sometimes tends to be "talked about" and not allowed "to speak" to the same extent. Hence, the aim of this study was to explore and compare how TGNB youth (16–26 years old) in Sweden and Spain experienced their access to healthcare, with a specific focus on challenges in gaining access. Such an analysis can shed light on strengths and limitations in different kinds of healthcare systems and illuminate context-specific challenges that need to be addressed to assure equitable access to healthcare for TGNB youth. The comparative approach can provide an understanding of how the context shapes healthcare access for TGNB youth and contribute valuable insights into improving healthcare provision and policy development.

## Theoretical framework

As already mentioned, access to healthcare is at the core of achieving equity in health. Accessibility is complex and is not merely about the physical difficulty of getting into the healthcare facility or measuring the distance between the care seeker's home and the healthcare facility [20]. Levesque et al. [20] define access as the opportunity to reach and receive appropriate medical care that matches a care-seeker's needs. Within this framework, access is the result of interaction between the characteristics of individuals, households, the social and physical environments, and the characteristics of the health system, organizations, and providers. Levesque et al. [20] identified five dimensions of access on the service side, and another five on the side of individuals, the community, and the population. In the present study, we were mainly interested in the healthcare service side, and so our analysis of the interviews used the concepts of approachability, acceptability, availability, affordability, and appropriateness.

Levesque et al. [20] mention norms, culture, and gender as important aspects of accessibility. In relation to this, we find it useful to employ the concept of cisnormativity, which can be described as the social discourses and practices that assume that individuals identify with the gender they were assigned at birth. Cisnormativity contributes to the construction of TGNB people as unintelligible, and can also be a part of institutional pathologizing discourses (see, for example, [3,21]). Using Levesque et al.'s conceptualization of accessibility together with the concept of cisnormativity allows us to explore how specific ideals and normative expectations might affect access to healthcare for TGNB people.

## Materials and methods

### Study contexts

Healthcare in Sweden is mainly the responsibility of the 20 regions, which have partially autonomous responsibility for providing healthcare. However, care for gender dysphoria is considered specialized care and is partly centralized. There are only six official teams in Sweden that evaluate the need for gender-affirming medical procedures, and even fewer clinics provide gender-affirming surgery. To get to the evaluating team, care-seekers often need a referral from a general practitioner and/or a psychiatrist. Special multidisciplinary teams, often located in psychiatric clinics or sexological centers, are responsible for the evaluation, for diagnosing gender dysphoria/transsexualism, and for referrals to medical procedures. The evaluations consist of psychological, physical, social, and psychiatric components. Sweden has a general healthcare insurance system that also covers gender-affirming medical procedures [22]. People over 18 years old can get access to legal gender affirmation and genital surgery [23], and in practice, one must have a transsexualism diagnosis and a certificate from the evaluating psychiatrist in order to get that approved.

Spain also has a national health system, but each autonomous community is responsible for providing the necessary health services in its territory. This means that there are 17 health

services, one corresponding to each community, as well as the National Institute of Health Management, which is in charge of the health systems of the two autonomous cities (Ceuta and Melilla). Most, but not all, these communities have developed protocols for the care of trans people. In some regions, healthcare for trans people can be centralized in gender units, or it can be provided (or supposedly provided) in any healthcare center. Other communities without these specific services (or even protocols) refer their trans users to other communities, with a diversity of care within the territory. All costs derived from the care of health needs and transition are free for users in the public system. In the Region of Murcia, a region in the South of Spain from where the participants were recruited, Law 8/2016 [24] allows adolescents to receive hormonal treatment (with the consent of their parents or legal tutor) and adults to receive legal gender affirmation and genital surgery. This law indicates the establishment of a regional protocol (not created until 2020) and that health care should be based on informed consent, gender self-determination (without the need for a psychological or psychiatric certificate), non-discrimination, and comprehensive assistance.

## Study design

This study was based on a qualitative analysis of 16 interviews with young TGNB people living in Sweden and 18 interviews with young TGNB people living in Spain. All interviews were conducted between January 2020 and March 2021.

An abductive approach was applied, and the Levesque framework was used as a "reference point" since the two materials were collected with slightly different research questions. In other words, instead of directly comparing the two materials against each other, the conceptual framework allowed for a "neutral point" with which we could compare the analysis of each material. In this way, we could identify how the two contexts differed in terms of different aspects of accessibility. The analysis focused mainly on commonalities and revealed some kind of universality across contexts regarding TGNB people's access to care and barriers to such access. We also found some differences, but these must be interpreted with caution, as the issues might have been illuminated differently if the interview guides had been more similar.

## Recruitment

**Swedish sample.**   To recruit participants, advertising material was sent to networks, social media groups, and associations for transgender people. A targeted advertisement on social media was also used. The material contained contact information that potential participants could use to register their interest in participating. Those who did this were then contacted, given information about the study, and asked if they wanted to be interviewed. Due to the COVID-19 pandemic, all interviews were conducted by video call, using the Zoom software application.

**Spanish sample.**   The process of recruiting participants began by contacting various LGBTIQ+ associations and associations for families of trans minors, and presenting the project to them. Two participants were recruited through the dissemination of the study by the associations, and the rest were recruited by snowball sampling. Most interviews were face-to-face, but a few were conducted via video call.

The recruitment stopped when new interviews showed similar patterns to the previous ones. In both contexts, the themes of the last interviews were remarkably consistent with those observed in earlier interviews, which indicated that further interviews were unlikely to yield any radically different insights in relation to the aims of the research projects. All interviews were recorded using a voice recorder and transcribed verbatim in the original language (Swedish and Spanish).

## The interviews

The interviews were semi-structured, followed an interview guide, and conducted in Swedish and Spanish, respectively, by the two authors, whom have previous experience researching TGNB people's health and access to healthcare. The interview guide for the Swedish project consisted of open questions on four themes: (1) experiences of mental health and its determinants; (2) experiences of gender-affirming healthcare; (3) experiences of other types of healthcare; and (4) suggestions for improvements to improve health and access to healthcare. The interview guide for the Spanish project consisted of open questions on five themes: (1) experience of social transition; (2) social and family support; (3) experiences of gender-affirming healthcare; (4) experiences of other types of healthcare; and (5) proposals for improving healthcare. In both projects, follow-up questions were asked based on the participants' narratives. The interviews lasted between 35 minutes and 2 hours in both contexts.

## Participants

**Swedish sample.**   The 16 participants were between the ages of 16 and 25, and three of them were under the age of 20. Nine identified as male or transmasculine, four as non-binary, and three as female or transfeminine. They lived in different parts of Sweden. Nine were studying (five of them at university), four were employed, and three were unemployed. Thirteen had met with a gender evaluation team, and two were waiting for their first visit. Eleven had undergone some type of gender-affirming care such as hormone therapy, surgery, or voice therapy.

**Spanish sample.**   The 18 participants were between the ages of 15 and 26. Five of them were under 18 (the legal age of adulthood in Spain), and the rest were aged 20 or over. All of them lived in the Region of Murcia. Thirteen identified as male or transmasculine, two as non-binary, and three as female or transfeminine. Four of them were in vocational training or had finished their training, ten were studying, and four were finishing their university degrees; three of them were studying degrees related to health (nursing, psychology, and medicine). Fourteen had undergone some type of gender-affirming care, and the remaining four were undergoing preparatory visits or waiting for their first visit and had not started hormonal treatment.

In the findings, the participants are referred to using pseudonyms in combination with "TM" for those identified as male or transmasculine, "NB" for those identified as non-binary, and "TF" for those identified as female or transfeminine. In addition, "SE" is used for participants from the Swedish material and "ES" for those from the Spanish material.

## Analysis

The analysis largely followed the steps proposed by Braun and Clarke [25], and involved getting to know the material by reading through it repeatedly, initial coding of the material, preliminary thematization of the material, and revision of themes in relation to the analytical framework. In the first step, we each read our respective materials and then picked out parts of our interviews that dealt with access to healthcare. In the second step, both authors read the parts from both contexts and discussed initial interpretations and thematization. Based on these preliminary analytical thoughts, we chose the conceptual framework. In the next step, we organized the codes and quotations in terms of the five different aspects of accessibility [20]. We then both analyzed each part in detail, and inductively identified sub-themes within those aspects of accessibility which were experienced as more limited.

### Ethical considerations

Before each interview began, the interviewer carefully reviewed the information letter and received written or verbal (recorded) consent from the participant. In the case of underage participants in the Spanish context, their parents or legal guardians signed the informed consent, and the adolescents signed the informed assent. In Sweden parents or guardians do not have to give consent if the participants are above 15 years old.

## Results

In presenting the findings we follow the structure of the accessibility framework, focusing on the five "A"s: approachability, acceptability, availability, affordability, and appropriateness. However, it was clear that many of the inductive themes were concerned with the appropriateness of the services.

### 1. Approachability

According to Levesque et al. [20], approachability comprises aspects such as information, screening, outreach, and transparency. These aspects were quite peripheral in both the Swedish and Spanish contexts. However, when Mika (NB, SE) was asked if they had been screened for HPV, they said that they had not gone to the appointment:

> I just tore it up. I also know that I'm much less at risk than others because I don't have sex, I don't do things. So I felt a bit safer being able to rip the invitation up. Sometimes I could have wished it would have been a lot easier if they could just do some sort of self-test. So that you can do it yourself.

Joachim (TM, SE) had also experienced problems with HPV screening, but considered himself lucky compared to others who were invited to the screening:

> And then you have to keep track of it yourself every three years. It's very easy to deliberately postpone it because it's so mentally difficult to think about. At the same time, I'm lucky, and it's because I, I've gotten information from the trans care clinic where they say, "OK, this clinic has experience with trans people and LGBTQ people." And what you can take from that is that we know they do a good job of not being ignorant when it comes to transgender people and cell sampling.

Joachim's experiences also illustrate the importance of knowing where to turn to get trans-competent healthcare. However, other participants had different experiences, feeling that such knowledge was not easily accessible and that they did not know where they could go to get trans-competent care.

### 2. Acceptability

According to Leveque et al. [20], acceptability concerns professional values, norms, culture, and gender. Overall, this theme was not prominent in the narratives in either of the contexts, but two issues did stand out, especially in the Swedish context. First, the participants' experiences showed that there was a lack of trust and a lack of willingness to seek care. Dylan (TM, SE) said:

> I usually turn to like adult psych because I still have a contact there. Plus I'm pretty scared of the healthcare center actually. /. . ./ Because I called them one day about a problem and I

*mentioned that I was trans so they were just "A-ha, so you're a woman then?" /. . ./ They might know how to inject [the testosterone shot] and so on. But I don't feel safe there, but it feels like the endocrinology clinic still has a little more knowledge about the testosterone part and the trans part really.*

Hence, some participants experienced that the parts of healthcare that dealt with gender-affirming care had more knowledge of trans issues, and they preferred to turn to those places instead of, for example, the healthcare center. In a similar vein, Joachim (TM, SE) said:

*I don't feel like I can trust every doctor I go to in the health center context, for example, when I hurt my foot. I don't feel like I can trust those doctors not to think that transgender people shouldn't exist for whatever reason. I can't trust mainstream healthcare to be trans positive.*

In a similar vein, Rosa (TF, ES) said: "*It's true that you could say that in the end, the medical issue, a doctor doesn't feel like a safe place, no, it's true.*" However, there were also experiences of a lack of trust in gender-affirming healthcare; for example, in terms of how non-binary people were treated or regarding the gatekeeping evaluations. We will return to this point later.

Some participants had been avoiding or postponing seeking care in an attempt to avoid bad experiences. Billie (NB, SE) said: "*I don't seek care if I don't really have to,*" and Kian (TM, SE) stated: "*If I hadn't been in such great need of care, in most cases I might have waited to seek care from the healthcare center, since I know what treatment I'll get there.*" Manuel (TM, ES) also explained: "*I don't want to go to the doctor, but that's just because I don't want to have to explain myself again, you know?*" In this way, previous experiences of healthcare affected the acceptability of healthcare and made some of the young trans people postpone or avoid seeking care, possibly resulting in adverse health effects.

## 3. Availability

Availability of services describes the possibility of accessing services both physically and in a timely manner [20]. A prevalent aspect of availability in both contexts was the issue of waiting for healthcare. In Sweden, care seekers generally need a referral to gain access to an evaluation for gender dysphoria. The participants' narratives suggested that it could be a winding road to get this referral, both in terms of waiting for appointments and due to being forced to undergo an extra evaluation, which caused further delay. For example, Billie (SE) and Antonio (ES) said:

*It ended up being like six months from when I started until I got into the queue, and it was at least 20 months of waiting time [until the first visit to the evaluation team]. (Billie, NB)*

*It was the first time I went to the family doctor, and I asked him to refer me to the endocrinologist for this reason, umm. . . I had to wait for the first appointment, it was seven months, seven months! I began to think that everything was always going to go slowly, and I thought "My God. . . When am I going to start, when?" and I was desperate. (Antonio, TM)*

This delay and distancing between medical appointments could negatively influence the perception of the care received, as the participants felt they required more continuous care. Manuel (TM, ES) explained:

*He gave me appointments. . . every two months, every three months, and it's true that at the beginning I needed more attention, right? I was beginning the transition. . . I needed a little more support.*

Faced with delays in care, some participants turned to references or searched for information outside the health system, for example, via LGBT associations or groups. Jaime (TM, ES) said:

*In the end, I see my endocrinologist once every six months, five. . . in the end, I have to look for myself a bit if I want to know something. . . so, in the end, my trans friends, I talk to them, I ask them things, we end up making a support network, or in LGBT groups or whatever, there are many people who know.*

Despite the delays, the participants still received their hormone therapy. However, surgery was not always available in the public hospitals of the Region of Murcia. Some of the participants had been referred to private hospitals or to other autonomous communities with gender unit services. This was the case for Alex (NB, ES) and Jaime (TM, ES), who perceived this referral to a gender unit as a positive thing, because it gave them access to trained health professionals and prevented them from feeling uncomfortable.

*In the end, he called me and said "Yes, maybe you can go to the gender unit at the Ramón y Cajal Hospital (Madrid) because they know about the subject there. . ." There, for example, they call you by your last name, or they ask you what you want to be called. That's much better and more regulated, that service is quite good. (Jaime)*

In terms of physical availability, some participants in both contexts described inadequate facilities. Alex (ES) and Antonio (ES) gave some examples regarding bathroom use:

*I ask myself "What bathroom do I use?" Because I'm also non-binary, and sometimes I read as feminine and sometimes as masculine. . . I just want to pee, but it seems that going to pee is a performance (laughs). (Alex, NB)*

*I'm looking for the bathroom, they tell me to go there, I go to the corridor, and I see the ladies' bathroom and the bathroom for disabled people, I keep walking. . . Then I remembered that I was in the breast unit (laughs). (Antonio, TM)*

Regarding waiting rooms for gynecological services, Alex (NB, ES) said:

*Going to the gynecologist in itself is uncomfortable for me, because many times I've been asked "Are you accompanying the person who just went in?" And I'm, "No. . . I'm waiting to go in myself." (laughs). They are stressful situations.*

Emilia (TF, SE) also referred to uncomfortable situations in the waiting room: "The only thing [with somatic healthcare] was someone who was surprised that I stepped forward when they called up 'Emilia' in the waiting room."

## 4. Affordability

Affordability concerns the capacity to spend economic and other resources to access services [20]. Both Sweden and Spain have tax-funded healthcare. However, our participants described experiences of paying or considering paying for gender-affirming care, for example in terms of meeting private psychologists, having private surgery, or ordering hormones from abroad. Given the long wait to receive mental health care or even inadequate treatment by

professionals, participants who could afford it might, for example, pay to go to a private psychologist. Alex (NB, ES) said:

*Before looking for a private therapist and getting well, I tried to go to psychiatry, and the answer I got from the psychiatrist. . . first he told me to take the pills he prescribed me and, when they stopped working and the adverse effects began, what he told me was "Well, you're going to be anxious for life, get used to the idea."*

In the Swedish context, affordability mainly played out in relation to the inaccessible care for gender dysphoria and the waiting times for gender-affirming care, which led to some care-seekers paying for hormones, laser treatment, or surgery to help them endure the waiting for tax-subsidized care. Sadie (TF, SE) said:

*And it got to a point where I was like, I can't, I literally can't [wait anymore]. I ended up self-medicating. I've ended up subsequently switching from that to GenderGP, so a informed consent-based thing online, and then also I've been out to my parents since December, and then [got financial help from them and] I went on by paying for a private diagnosis and laser treatment.*

Sadie's experiences also illustrate how the capacity to pay for care could be dependent on a person's financial situation and, as in this case, their relationship with their parents. Similar to Sadie, Charlie (NB, SE) could not stand waiting for surgery anymore. After waiting for 3.5 years, Charlie decided: "I gave up. And started to eat oatmeal and save money to pay for it myself." Charlie also ended up paying for a mastectomy within private care.

## 5. Appropriateness

As already mentioned, the appropriateness of healthcare services was a major concern among the participants in both contexts. Appropriateness is described by Leveque et al. [20] as the fit between the care-seekers' needs and the services, and the interpersonal quality of the services. The issues mentioned by the participants ranged from being referred to by the wrong name and/or pronoun, normative assumptions, overt discrimination, meeting care providers who lacked knowledge, having to take responsibility for knowledge gaps, and encountering gatekeeping.

**5.1 Wrong name and pronouns.** Being referred to with the wrong name and pronoun came up in both contexts. Alex (NB, ES) described how exhausting it could be when the wrong name was used over and over again:

*Above all, before changing my registered name, I was constantly referred to by my deadname. If I went to the health center or I went to the hospital, it meant correcting people all the time, and there comes a time when you get tired, you stop doing it because no one listens to you.*

Charlie (NB, SE) also did not have good experiences of correcting people: "*I've always been misgendered by health professionals, and also criticized when I've corrected people.*" While in some cases the use of the wrong pronoun and name was connected to cisnormative assumptions, in other cases it continued even after the participants informed their healthcare providers of the correct name and pronoun. Susana (TF, ES) said: "*You tell them (health professionals) something and they always question you, and. . . that bothers you a lot, a lot. [. . .] She addressed me in the masculine, even having said it and even knowing it, so. . . it seems a bit unprofessional to me.*"

The issue of using the wrong name could also become relevant in the waiting room, as Kian (TM, SE) described:

*Right now I have a health center that's very confused about names, which I notice all too often. They'll call "Kian", and when I'm about to stand up, they turn and ask, "Karin?" And I say, "No, Kian."*

In the case of Joachim (TM, SE), the healthcare system had documented a gender-neutral pronoun in his medical record, but as he explained, this was not correct: "*I use the pronoun 'he' /../ and they hadn't dared to ask me. So it kind of continued like that until we'd met five times and then the caregiver asked, 'But wait, a-ha, you identify as a man? I hadn't understood that from your journal.'*" Hence, although the incorrect pronoun was probably documented with good intentions, the lack of routines for asking about and documenting pronouns contributed to misgendering.

One of the participants from the Spanish context commented on the administrative difficulties in being addressed with the appropriate name and pronoun, as it was necessary not only to change one's name on the national identity document, but also to change it specifically in the healthcare system: "*There's one thing that many trans patients don't know, and that's that in the hospital, when you change your name. . . if you don't go to the hospital to change your data, the old data will continue to appear, so they'll keep calling you by the old name.*" (Juan, TM ES)

**5.2 Normative assumptions and "ignorant benevolence".**   Several participants described how even though the healthcare professionals might not have bad intentions, there could still be problematic aspects or issues of lack of knowledge in the encounter. Connor (TM, SE) described this as "*ignorance*", while Joachim (TM, SE) said: "*I feel that the most common thing from the healthcare staff is ignorant benevolence,*" giving an example of how "*the nurse had to double check, 'So you're a man, but it says woman here?'*" Joachim explained that this was typical for his healthcare encounters and meant that he had to "*keep the best eye on myself /. . ./ But I haven't come across anyone who hasn't wanted me to be well.*" Antonio (TM, ES) gave an example where a healthcare provider had deduced that his gynecomastia was due to a medical condition rather than thinking that he was transgender.

*. . .asked me why I'd had a mastectomy. . . I don't think that she did it with bad intention, but at that moment I couldn't answer her. . . and she said. . . I don't remember exactly, if I had inherited it from my father or my mother, then I automatically realized that she thought I was a cis guy with gynecomastia, and I realized that, of course, she didn't think I was trans, and I didn't get to correct her.*

This issue of non-optimal outcomes despite healthcare providers having good intentions could have specific effects for non-binary trans people:

*So it's also been a problem that I'm non-binary. Because there [at the youth clinic] they've been like, "It's totally OK to say you're a guy." And then it'll be like, "Well, but I've just said I'm not." So there it's almost felt like they're also trying to push you into a category, even if they're trying to be tolerant and so on, because you're an LGBTQ person, they'd still prefer you to be a certain type of LGBTQ person. (Charlie, NB, SE)*

*During the transition and going from realizing that I'm a man to realizing that I'm a non-binary person, I didn't tell my doctors because I knew that the consequences could be fatal. I'll save it, that silly discussion, because I know they're not going to react well. (Alex,NB, ES)*

Normative assumptions recurred in the material; for example, how healthcare providers made assumptions based on the person's gender assigned at birth or their legal gender identity. The issue of "coming out" could be connected to cisnormative ideas; not only building on the assumption that everyone is cis until they come out, but also, as the quotation below from Thom (TM, SE) illustrates, the uncertainty about what reactions one will get when doing so.

*The problem is when you get a new care contact. It's very tough. /. . ./ That's fucking hard, sometimes. When you get a new contact. And you kind of have to come out. Say it and get it established and see people's reactions to it.*

The issues of ignorance and normative assumptions were experienced as distracting attention from the medical problem for which the participants were seeking care. Kian (SE) and Pedro (ES) said:

*There's that focus on their confusion about my gender identity again, and I have to explain it to them instead of actually getting help with what I'm looking for. For example, depression or stuff like that. (Kian, TM)*

*I don't understand why you have to ask me if it was from a girl to a boy or from a boy to a girl. (Pedro, TM)*

Some participants decided to avoid mentioning their gender or trans identity, in order to avoid distracting attention from their medical consultation or to avoid having to give explanations. Jaime (TM, ES) said:

*Unless it's strictly necessary that. . . for example, there was a time when they had to give me some medication. . . I told him "Hey, but I take testosterone, is this going to affect that?" then maybe someone asks you, and then you tell him, "Well, I take testosterone for this." But that, unless it's strictly necessary, I don't say anything about that.*

**5.3 Overt discrimination.**   While many of the bad experiences of healthcare were said not to derive from bad intentions, some of the participants' experiences were of more overt or explicit discrimination. Francisco (TM, ES) said:

*When I had the appointment with the surgeon from the national health system, he was looking at my chest to see what surgery I would have to have and so on. . . and at that moment he told me that. . . that I had very nice breasts. . . and I was like "What?" Obviously, I left, I left there and had the operation privately, I said that I wasn't going back there, no way. I left the consultation crying.*

When Charlie (NB, SE) was asked if they thought it mattered that they were non-binary when meeting a healthcare provider, they said:

*All my contacts at general psych have written in the journal that I "claim to be non-binary." And so it says this in the assessment, as well as, "21-year-old young woman who claims to be transgender."*

Charlie's experiences show not only that it might be more difficult for non-binary people to be taken seriously, but also a normative idea that this is "a claim" rather than a reality.

**5.4 Lack of knowledge among care providers.**   One recurring issue among participants from both contexts was a lack of knowledge among their care providers. This lack of knowledge included somatic issues such as side effects from long-term hormone treatment, knowledge about what it was like to live in a cisnormative context, what it meant to be trans, and how gender-affirming healthcare worked.

*When I called my doctor, that was it, because the first step is to call the doctor and for her to give you an appointment with the endocrinologist, when I spoke with her, which was not very much, but it's true that you could tell that she didn't understand the subject. In fact, she told me "Well, if you know more about this than I do," which seems to me to be at least correct because it tells you the truth. (Rosa, TF,ES)*

Thom (TM, SE) gave another example regarding cervical cancer screening:

*Like this week when I was going to book for the HPV cell sampling. Then I got called up and they kind of got like, "No, but I don't even know how to. . . Is it for you or is it for someone else, or? I actually don't know how to take a sample from a man."*

This lack of knowledge among care providers sometimes meant that the participants had to educate their healthcare providers. Juan (TM, ES) gave an example:

*When I went to the family doctor, because he was the one who had to refer me to the endocrinologist, he didn't know what he had to do. I knew because before I went to the doctor I was watching documentaries, looking for information. . . so I knew the steps I had to follow, but I think he had no idea.*

In a similar vein, Cruz (ES) stated:

*So many times you go to the family center, to the family doctor, and they ask you "OK, and what do you want me to do with you, where do I have to refer you, what's the protocol, what's supposed to be the next step?" You have to go there knowing your legal rights, being informed, and knowing exactly what you want, and it's a responsibility that doesn't get placed on other people wanting healthcare.*

This shows the responsibility that the care-seekers took on, and also the knowledge that they had acquired to navigate the chain of care. The lack of knowledge among care providers meant that these trans care-seekers had to take responsibility for their care in a way that others might not have to do to the same extent. However, in some cases, the healthcare providers had taken responsibility for their lack of knowledge. Emilia (TF, ES) said:

*Then the other psychologist that I had later, she wasn't knowledgeable either, but she could somehow empathize and she was still willing to learn somehow, so that there I still felt like I could actually bring things up and discuss them with her.*

Some participants who were active in trans associations had offered to train professionals, but had encountered barriers, as Pedro (TM, ES) explained:

*It's not being wanted, then when you find an institutional refusal with this type of thing, it happens a bit like with schools. . .. That's when you realize where the real barrier is. That it's*

*no longer so much that each of the health personnel has to take an individual action to include the group, but rather that there's an entire organization that has to be organized for that.*

Pedro illustrates an important point here; that even if several of the participants experienced a lack of knowledge among individual providers, this was not happening in a vacuum, but could rather be seen as an expression of the limitations of the healthcare system as an organization and as a whole.

**5.5 Encountering gatekeeping within care for gender dysphoria.**  Gaining access to gender-affirming medical procedures was important for many of the participants. However, navigating the access to such care was experienced as difficult in many ways, in both contexts. In the Region of Murcia, Spain, prior to the implementation of the new protocol [26], it was mandatory to have a positive psychological evaluation in order to access hormonal treatment. This evaluation made some participants uncomfortable since they considered that it pathologized their gender identity. Some of them also perceived that it made the professionals uncomfortable:

*In the psychological evaluation, being in front of the psychologist who conducted it, and she was asking me some questions that were embarrassing, but she knew perfectly well that the questions she was asking were embarrassing. . . well, she looked at me with a face of "sorry" /. . ./ because, of course, the poor girl was asking me if I was wearing underpants or panties."* (Pedro, TM, ES)

This shows not only the humiliating questions that people with gender dysphoria have to answer in order to get access to gender-affirming care, but also how individual healthcare providers might struggle within a healthcare system that has put up certain standards for providing access to care. In order to negotiate and navigate access to gender-affirming care, some participants talked about a more strategic approach in the encounter with healthcare. Dylan (TM, SE) said:

*I skipped talking about how I didn't feel binary. There's a lot I leave out really just because I want to. . . It sounds like I've lied my way around. I don't mean I lie, but of course, I skip talking about certain things.*

In a similar vein, Eddie (TM, SE) stated: "*I introduced myself as a trans man, very binary, because I was worried that if I showed any other side [of myself], I might not get hormones.*" Along with a binary/non-binary gender identity, other factors described as affecting accessibility were age and mental health. Timo (TM, SE) said: "*At first I was too young to get a referral, then I was too old to get a referral, then I was too autistic to get a referral, and then I was suffering too much to get a referral.*" Another aspect that came up was being evaluated as not being "trans enough" in relation to gendered norms. When Emilia (TF, SE) tried to get a referral to the specialized team for gender dysphoria, the doctor asked "*lots of questions around gender stereotypes that I couldn't answer 'yes' to, and it felt, it was a bit like this fear, like, I won't get the referral.*" Alex (NB, ES) mentioned that his unwillingness to have a hysterectomy or phalloplasty, in order to retain his reproductive capabilities, made the endocrinologist question his identity:

*He sent me back to psychiatry, because I wasn't man enough for them, apparently, and well, [and] they pressured me because according to them, testosterone could increase the risk of*

*tumors in the uterus. . . which today I know is not only a lie, but also that there's no study that's tested whether that's true or not, it doesn't exist.*

These obstacles, which are often conceptualized as healthcare professionals gatekeeping access to gender-affirming medical procedures, might constitute an absolute barrier in getting access to care, but also in some cases might contribute to delays, or the care-seekers having to put in a lot of work to move forward in the chain of care. Juan (TM, ES) said that the endocrinologist "lengthened the tests that they had to do" but also that there were problems in the collaboration between different professionals and clinics:

*They refer you to a psychologist, the psychologist talks to you and makes a report, you take it to the endocrinologist, and he tells you that this report isn't valid. . . They refer you back to the psychologist, the psychologist is the same, and he doesn't know what to put in the report if the first one isn't valid. They don't deny you assistance, but they delay it.*

According to Antonio (TM, ES), such problems might be connected to a lack of knowledge among some care providers: "*They wouldn't know what to do, so they pass the ball from one to another, go there, the other one sends it to another place.*" Charlie (NB, SE) reported similar experiences in the Swedish context:

*At first it took like three and a half years to send referrals without me ever being able to get there [to the evaluation]. Because quite a few of the referrals weren't accepted, instead they were like, "Ah, no, but we also need a referral from. . . [specialized psychiatry]."*

In this sense, the access to gender-affirming care was often delayed, sometimes due to lack of knowledge in the chain of care and sometimes due to standards for access (gender stereotypes, binary norms, mental health) that some participants might have found it more difficult to meet.

## Discussion

Our analysis of 34 young people's experiences of access to healthcare in Spain and Sweden revealed several barriers that were similar between the two countries. This points to issues that exist independently of the specificities in the contexts and the differences between these two healthcare systems. While barriers existed in all stages of access, it was evident that the major barriers arose in the last step of access; that is, the appropriateness of services. This could suggest these two publicly funded subsidized healthcare systems have lowered (but not eliminated) the barriers concerning, for example, availability and affordability.

These results show that social position can affect the accessibility of healthcare, since only those with more financial resources can afford to pay for a service they need when the waiting time in public health is very long or when they want to avoid discriminatory experiences. The barriers regarding the first stages of access may be accentuated when it comes to intersectional categories such as migration status and, in specific cases, also age; this was not completely captured in our materials. Not having a residence permit can hinder access in a very early stage of accessibility, for example in terms of legal restrictions on accessing healthcare that is not acutely needed [27]. This lack of healthcare access is especially serious, given that this population may experience poor health and wellbeing and a limited ability to exercise power due to intersectional discrimination involving ageism, cisnormativity, and xenophobia [10].

Healthcare access can also be especially problematic for trans youth [16]. When it comes to age, especially in recent guidelines for care for gender dysphoria in the Swedish context, those

below 18 years old have more restricted access to care for gender dysphoria than their older counterparts [19]. Our analysis suggests that a non-binary gender identity can affect access in several ways, illustrating how cisnormativity influences access to healthcare in terms of conceptualizing gender as dichotomous and binary [28].

There were also some results that could point to differences between the contexts. In the Swedish context, care for gender dysphoria constitutes a very specific part of the healthcare that is centralized and, in many cases, separated from other kinds of care. Centralized care and access can lead to geographic inequities and loss of income due to the need to travel, which might have a larger effect on those with a low social position. Likewise, there were tendencies for the Swedish participants to talk very separately about care for gender dysphoria and other healthcare, while the participants from Spain saw healthcare as more unified.

In a recent cross-country comparison of care for gender dysphoria across European countries, the European Professional Association for Transgender Health [29] concluded that Sweden was one of the least well-performing countries in terms of accessibility of gender-affirming care. Spain, on the other hand, was ranked second among the countries compared.

It is clear in both contexts how the community has developed strategies to counteract limited access to healthcare, for example in terms of acquiring knowledge of care protocols and finding ways to negotiate access to gender-confirming care; but also, as a community, offering to educate healthcare providers. These responsibilities are taken on both by individuals themselves and by the community at large, for example by having platforms to share experiences and strategies to access healthcare. This can be understood as a responsibility shift, in which the care-seekers have knowledge and take on responsibilities that belong among the care professionals [6,12]. Moreover, these responsibilities might include tasks that the more privileged members of the population either do not have to do to the same extent or are well-equipped to do. The responsibility shift can be seen in many "stages" of access, from issues of knowing where to seek competent care to knowing where referrals should be sent.

A report by Transgender Europe [8] showed that individuals with bad experiences from healthcare were more likely to postpone future healthcare visits than those without such experiences. This shows how appropriateness feeds back into approachability and acceptability. Hence, the serious concerns raised about appropriateness can have detrimental effects on overall accessibility and in the end the utilization of healthcare. From an equity perspective, and in relation to the fact that TGNB people tend to report more, for example, mental ill health than the average in the general population, this is alarming. Optimally, a healthcare system should work to reduce inequities in health by providing more healthcare for those in greater need [1]. However, our results point in the opposite situation; that is, there might be a risk that health inequities are increasing in relation to healthcare because access is experienced as being limited in several ways.

## Limitations

As the materials were not gathered with the same aim or with the same interview guide, there are clear limitations in our comparative approach. However, we found the partly deductive (i.e. abductive) approach useful in order to put our analysis in relation to the framework, instead of putting the two materials directly in relation to each other. This allowed us to analyze in depth the different aspects of accessibility in relation to the two different materials, and then compare the results. The two authors independently analysed the selected parts from the two materials and compared the findings to enhance the rigour of the analysis.

The use of different interview guides might have made us miss some aspects that were more prevalent in one of the contexts; for example, the issue of screening only came up in the

Swedish material while the issue of bathrooms in healthcare services only came up in the Spanish material. Such aspects need to be further explored. Even though the two healthcare systems are different, they also share similarities regarding the availability of tax-funded access to gender-affirming care. Hence, there is also a need to further explore specific barriers to access in healthcare systems where access depends on private healthcare insurance or private means.

Although the recruitments aimed for a diversity of participants, the study populations were quite homogenous when it came to ethnicity, and few had migrant background. Hence, intersections with racism in healthcare and issues of access for undocumented migrants remain to be further explored.

The use of video calls for data collection might have had negative effects on possibilities to follow-up on non-verbal communication. However, we experienced that the interviews were as deep and detailed as the face-to-face interviews and interviews in previous projects. The video calls also made it possible for people that do not want to meet face-to-face to participate in the study. The data collection was carried out in the beginning of the pandemic and very few interviews touched upon the COVID-19. Therefore, the identified limited accessibility does not reflect the pandemic situation.

## Conclusions and implications for practice

Young transgender and non-binary people experience barriers in accessing healthcare both in the Spanish and the Swedish context. While barriers exist in all stages of access, it is evident that the major barriers arise in the last step of access; that is, the appropriateness of services.

To provide comprehensive quality care to TGNB people, it is necessary to train health professionals, improve collaboration between professionals and services, adapt physical spaces to make them more inclusive, and reduce waiting times. The strategies implemented to reduce barriers to healthcare access must be framed within the critique of, and resistance to, cisnormativity, and should focus on users with intersecting marginalized identities (children/youth, non-binary people, racialized people, and migrants) to promote health equity.

## Author Contributions

**Conceptualization:** María del Mar Pastor Bravo, Ida Linander.

**Data curation:** María del Mar Pastor Bravo, Ida Linander.

**Formal analysis:** María del Mar Pastor Bravo, Ida Linander.

**Funding acquisition:** Ida Linander.

**Methodology:** María del Mar Pastor Bravo, Ida Linander.

**Writing – original draft:** María del Mar Pastor Bravo, Ida Linander.

**Writing – review & editing:** María del Mar Pastor Bravo, Ida Linander.

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
