## [Decision Letter · Decision Letter 0]

14 Feb 2024

PONE-D-23-29925Access to healthcare among transgender and non-binary youth in Sweden and Spain: A qualitative analysis and comparisonPLOS ONE

Dear Dr. Linander,

Thank you for submitting your manuscript to PLOS ONE. After careful consideration, we feel that it has merit but does not fully meet PLOS ONE’s publication criteria as it currently stands. Therefore, we invite you to submit a revised version of the manuscript that addresses the points raised during the review process.

We look forward to receiving your revised manuscript.

Kind regards,

Julia Morgan

Academic Editor

PLOS ONE

Reviewers' comments:

Reviewer's Responses to Questions

**Comments to the Author**

1. Is the manuscript technically sound, and do the data support the conclusions?

Reviewer #1: Yes

Reviewer #2: Yes

2. Has the statistical analysis been performed appropriately and rigorously? 

Reviewer #1: N/A

Reviewer #2: N/A

3. Have the authors made all data underlying the findings in their manuscript fully available?

Reviewer #1: No

Reviewer #2: No

4. Is the manuscript presented in an intelligible fashion and written in standard English?

Reviewer #1: Yes

Reviewer #2: Yes

5. Review Comments to the Author

Reviewer #1: This is an interesting paper and the emphasis on comparing analyses between the two countries rather than the data is well-made in the text, and it strengthens the findings. There are many strengths and the authors draw on key literature. Perhaps a couple of points for improvement:

1. The importance of the study simply lies on the fact that young people face exacerbated difficulties in relation to healthcare access. This is not offering much to the reader or the body of literature. It would strengthen the thesis if a robust reasoning for the study is offered and integrated in the abstract.

2. Perhaps some additional input on what the implications of this new knowledge are would help appreciate the conclusions further.

3. On page seven, the claim is made for saturation of the data. This is a rather bold claim and if so, much more detail is needed of how the researchers considered that the data reached the point of full representation of this part of the populations in Spain and Sweden, respectively.

4. How were the findings validated, if any method was used?

5. Some demographics were collected and specifically participants identified either as male or trans masculine; non-binary; or female or trans feminine. Was there any analysis about differences within the group and their experiences? Non-binary is highlighted in the discussion but possibly adding simply those identifications in the extracts from the interviews in the text would support the reader's understanding of those experiences. This would not interfere with ensuring anonymity and privacy of the information.

Overall, the text could provide more legal-specific contexts of the two countries, too, which helps contextualise the findings, given the international audience of the journal.

Reviewer #2: Thank you for the well written and researched paper on an important topic. I have a few comments:

1. Please defined "cisnormativity" in the introduction.

2. In the methods, when describing the two countries, make the differences more clear, and provide references.

3. Given data collection took place during COVID-19, some more thought in the discussion how this may have impacted data collection and participants' responses, and whether the experiences they described were impacted by COVID-19.

4. More rationale and detail about why some interviews still took place in-person, and some of the ethical issues around having in-person data collection during COVID-19.

5. It is unclear, were the interviews conducted in Spanish and Swedish, and was the transcribing and analysis done in the original language (Spanish or Swedish)? At what stages were the findings combined from the two countries, and what language was used for this?

6. It is unclear, did each of the authors analyze their own interviews? How was calibration across interviewers and analysts maintained?

7. More about reflexivity, the interviewers' backgrounds, and how rigour was maintained in the analysis is needed.

6. PLOS authors have the option to publish the peer review history of their article (what does this mean?). If published, this will include your full peer review and any attached files.

Reviewer #1: No

Reviewer #2: No

---

## [Author Response · Author response to Decision Letter 0]

14 Apr 2024

Dear editor and reviewers, 

Thank you for the opportunity to send in a revised version of our manuscript. We would like to thank the reviewers for their productive comments regarding our manuscript; we have found them useful and believe they have helped us to improve our manuscript. We have now prepared a revised manuscript in which we have addressed the comments as shown below. 

We hope that with these changes you will accept our manuscript for publication, and we look forward to hearing from you. 

Bold = comments from editor or reviewers. 

Normal font = Our response to the reviewers’ comments.

Editor comments

1. Please ensure that your manuscript meets PLOS ONE's style requirements, including those for file naming. ThePLOS ONE style templates can be found at https://journals.plos.org/plosone/s/file?id=wjVg/PLOSOne_formatting_sample_main_body.pdf and https://journals.plos.org/plosone/s/file?id=ba62/PLOSOne_formatting_sample_title_authors_affiliations.pdf

Thank you for noticing this, we have changed so that they match. 

3. Your ethics statement should only appear in the Methods section of your manuscript. If your ethics statement is written in any section besides the Methods, please move it to the Methods section and delete it from any othersection. Please ensure that your ethics statement is included in your manuscript, as the ethics statement enteredinto the online submission form will not be published alongside your manuscript. 

All the ethic statement appers now in the Method section and was deleted from the end of the manuscript.

4. Please review your reference list to ensure that it is complete and correct. If you have cited papers that have been retracted, please include the rationale for doing so in the manuscript text, or remove these references and replace them with relevant current references. Anychanges to the reference list should be mentioned in the rebuttal letter that accompanies your revised manuscript. If you need to cite a retracted article, indicate the article’s retracted status in the References list and also include a citation and full reference for the retraction notice.

There are no retracted papers cited on the reference list. We have added some references to accomplish the suggestion of reviews and updated the reference list.

Reviewer #1: 

1. The importance of the study simply lies on the fact that young people face exacerbated difficulties in relation to healthcare access. This is not offering much to the reader or the body of literature. It would strengthen the thesis if a robust reasoning for the study is offered and integrated in the abstract.

We have elaborated on this in the introduction and revised parts of the abstract. 

2. Perhaps some additional input on what the implications of this new knowledge are would help appreciate the conclusions further.

We have added some implications after the aim in the introduction. However, we have kept it short due to the word-limit. 

3. On page seven, the claim is made for saturation of the data. This is a rather bold claim and if so, much more detail is needed of how the researchers considered that the data reached the point of full representation of this partof the populations in Spain and Sweden, respectively.

We have elborated on this part, the paragraph now reads: 

“The recruitment stopped when new interviews showed similar patterns to the previous ones. In both contexts, the themes of the last interviews were remarkably consistent with those observed in earlier interviews, which indicated that further interviews were unlikely to yield any radically different insights in relation to the aims of the research projects.”

4. How were the findings validated, if any method was used?

In neither of the contexts, the research group did member checking or in other ways validated the findings. The transcriptions were made verbatim and were analyzed both in the original language by respective author so as not to lose nuances of understanding but also by both authors in English. The quotes shown in the results are derived from them.

The analysis was carried out largely following the steps proposed by Braun and Clarke and in relation to the the theoretical framework in an abductive manner. 

5. Some demographics were collected and specifically participants identified either as male or trans masculine;non-binary; or female or trans feminine. Was there any analysis about differences within the group and their experiences? Non-binary is highlighted in the discussion but possibly adding simply those identifications in theextracts from the interviews in the text would support the reader's understanding of those experiences. This wouldnot interfere with ensuring anonymity and privacy of the information.

We have added information about gender identity next to each quote as suggested. However, we do not find it possible to draw any firm conclusions about differences between different groups. There were no such clear trends in the material. 

6. Overall, the text could provide more legal-specific contexts of the two countries, too, which helps contextualise the findings, given the international audience of the journal.

When describing the contexts we have added central legal aspects connected to the issues explored, see p. 5-6. 

Reviewer #2: 

1. Please defined "cisnormativity" in the introduction.

We have defined “cisnormativity” in the Theoretical framework, the first time, that it appears in the text.

2. In the methods, when describing the two countries, make the differences more clear, and provide references.

We have elaborated on the different contexts and added references. See. p. 5-6. 

3. Given data collection took place during COVID-19, some more thought in the discussion how this may have impacted data collection and participants' responses, and whether the experiences they described were impacted by COVID-19.

To our surprise, the video calls did work very well, and compared to our previous data collections, the interviews were as deep and detailed as comparable interviews we have done before, face-to-face. However, of course there might be a risk of missing non-verbal communication which could have facilitated for example, more follow-up questions etc. One reflection was that some individuals that might not have participated in face-to-face interviews signed up for participation when it was video calls (some without video). However, this we do not know for sure, but people that were a bit socially isolated participated. The data-collection in Sweden was carried out just in the beginning of the pandemic and in Spain little before are at the begining of the pandemic and not many of the interviews touched upon the COVID-19. We have added a reflection in the limitations section (see p. 29). 

4. More rationale and detail about why some interviews still took place in-person, and some of the ethical issues around having in-person data collection during COVID-19.

The data-collection in Spain was carried out before and just at the beginning of the pandemic, that's why the latest interview was made by video calls per preference of the interviewees, even without the mobility restriction measures by law having yet been implemented.

5. It is unclear, were the interviews conducted in Spanish and Swedish, and was the transcribing and analysis done in the original language (Spanish or Swedish)? At what stages were the findings combined from the two countries,and what language was used for this?

This information has been added to the methods section, see p. 8-10.

6. It is unclear, did each of the authors analyze their own interviews? How was calibration across interviewers and analysts maintained?

This has been clarified in the analysis section on p. 9-10. 

7. More about reflexivity, the interviewers' backgrounds, and how rigour was maintained in the analysis is needed. 

We have added previous research experiences on p. 8. And we have elaborated on rigour in the limitations part on p. 28.

---

## [Editor Report · Decision Letter 1]

24 Apr 2024

Access to healthcare among transgender and non-binary youth in Sweden and Spain: A qualitative analysis and comparison

PONE-D-23-29925R1

Dear Dr. Linander,

We’re pleased to inform you that your manuscript has been judged scientifically suitable for publication and will be formally accepted for publication once it meets all outstanding technical requirements.

Kind regards,

Julia Morgan

Academic Editor

PLOS ONE
---

## [Editor Report · Acceptance letter]

30 Apr 2024

PONE-D-23-29925R1 

PLOS ONE

Dear Dr. Linander, 

I'm pleased to inform you that your manuscript has been deemed suitable for publication in PLOS ONE. Congratulations! Your manuscript is now being handed over to our production team.

Kind regards, 

on behalf of

Dr. Julia Morgan 

Academic Editor

PLOS ONE